# The Impact of New Treatments on Short- and MID-Term Outcomes in Bilateral Lung Transplant: A Propensity Score Study

**DOI:** 10.3390/jcm11195859

**Published:** 2022-10-03

**Authors:** Annalisa Boscolo, Andrea Dell’Amore, Tommaso Pettenuzzo, Nicolò Sella, Alessandro De Cassai, Elisa Pistollato, Nicola Cacco, Andrea Manzan, Agnese De Carolis, Federico Geraldini, Giulia Lorenzoni, Federica Pezzuto, Giovanni Zambello, Marco Schiavon, Fiorella Calabrese, Dario Gregori, Emanuele Cozzi, Federico Rea, Paolo Navalesi

**Affiliations:** 1Institute of Anesthesia and Intensive Care, Padua University Hospital, 267 C. Battisti, 35128 Padua, Italy; 2Thoracic Surgery Unit, Department of Cardiac, Thoracic, Vascular Sciences, and Public Health, University of Padua, 35128 Padua, Italy; 3Department of Medicine (DIMED), University of Padua, 35128 Padua, Italy; 4Unit of Biostatistics, Epidemiology and Public Health, Department of Cardiac, Thoracic, Vascular Sciences, and Public Health, University of Padua, 35128 Padua, Italy; 5Department of Cardiac, Thoracic, Vascular Sciences, and Public Health, University of Padua, 35128 Padua, Italy; 6Transplantation Immunology Unit, Padua University Hospital, 35128 Padua, Italy

**Keywords:** lung transplant, transplantation, ECMO, VA ECMO, tacrolimus, intensive care

## Abstract

Background: Despite many efforts to improve organ preservation and recipient survival, overall lung transplant (LT) mortality is still high. We aimed to investigate the impact of ‘prophylactic’ veno-arterial extracorporeal membrane oxygenation (VA ECMO) and tacrolimus on 72-h primary graft dysfunction (PGD) and 30-day acute cellular rejection, respectively. Methods: All consecutive LT patients admitted to the Intensive Care Unit of the Padua University Hospital (February, 2016–2022) were screened. Only adult patients undergoing first bilateral LT and not requiring cardio-pulmonary bypass, invasive mechanical ventilation, and/or ECMO before LT, were included. A propensity score weighting analysis was employed to account for the non-random allocation of the subjects to different treatments. Results: A total of 128 LT recipients were enrolled. Compared to the ‘off-pump’-group (n.47, 37%), ‘prophylactic’ VA ECMO (n.51,40%) recorded similar 72-h PGD values, perioperative blood products and lower acute kidney dysfunction. Compared with cyclosporine (n.86, 67%), tacrolimus (n.42, 33%) recorded a lower risk of 30-day cellular rejection, kidney dysfunction, and bacteria isolation. Conclusions: ‘Prophylactic’ VA ECMO recorded 72-h PGD values comparable to the ‘off-pump’-group; while tacrolimus showed a lower incidence of 30-day acute cellular rejection.

## 1. Introduction

The five-year survival rate for patients receiving LT is estimated to be around 60%, a lower rate than for all other solid organ transplants [1,2]. In fact, short and long-term outcomes after LT are critically affected by a complex interplay between donor and recipient-related conditions and risk factors. The in-depth knowledge of all of these contributors is mandatory for optimizing the intra- and perioperative management of LT.

In particular, many efforts have been made to clarify the applicability and benefits of intraoperative (io) veno-arterial extracorporeal membrane oxygenation (VA ECMO), using it not only as a ‘rescue’ device helpful for managing acute life-threatening complications (i.e., pulmonary hypertension with right ventricular failure) but also as an io extracorporeal support routinely applicable ‘in advance’ for every patient [3,4,5,6].

Indeed, io VA ECMO has been shown to minimize right heart strain during arterial clamping and to guarantee both lung protective ventilation strategies at low driving pressure and more controlled reperfusion of grafts reducing the risk of primary graft dysfunction (PGD) within 72 h after surgery [7,8]. On the contrary, conflicting data have been reported about the incidence of acute kidney injury (AKI) and the risk of bleeding and thromboembolic events [3,9,10,11,12].

Furthermore, the choice of the best immunosuppressive therapy after LT remains a matter of controversy [1,2]. A broad consensus exists about the use of a multidrug immunosuppressive therapy based on the association of corticosteroids, calcineurin inhibitors (usually cyclosporine or tacrolimus), cell cycle inhibitors, and (optionally) induction agents [1,13,14]. However, the question concerning which calcineurin inhibitor is best to administer is still under investigation [13,14]. Tacrolimus seems to reduce the incidence of organ rejection, bronchiolitis obliterans syndrome, nephrotoxicity, new-onset diabetes and malignancy as compared to cyclosporine, but no significant differences have been reported about short- and mid-term survival [1,13,14,15].

We hypothesize that the use of io ‘prophylactic’ VA ECMO and the adoption of a tacrolimus-based postoperative (po) immunosuppressive protocol are associated with lower 72-h PGD values and lower incidence of 30-day acute cellular rejection after LT (primary outcomes), respectively.

## 2. Materials & Methods

### 2.1. Study Population

All consecutive adult patients who received bilateral LT at the Padua University Hospital, from 1 February 2016 to 28 February 2022, and who were admitted to the post-surgical intensive care unit (ICU) were retrospectively screened. The exclusion criteria were: age < 18 years old, lung re-transplantation, need of io cardiopulmonary bypass (CPB), invasive mechanical ventilation, and/or ECMO before surgery or monolateral LT.

The study was approved by the Institutional Ethical Committee of Padua (reference number: 4539/AO/18); given the retrospective nature of the study, the need for a formal informed consent was waived. This article was written in accordance with the “Strengthening the reporting of observational studies in epidemiology” reporting checklist (Appendix A) [16].

Standardized protocols for io VA ECMO and immunosuppressive therapy have been developed in our center following international recommendations [1,4,5,13,14,17,18].

Patients were divided into three groups based on io support: (i) ‘off-pump’, if no extracorporeal support was adopted during surgery; (ii) io ‘prophylactic’ VA ECMO, if VA ECMO was applied from the beginning of the surgical procedure; and (iii) ‘rescue’ VA ECMO, if extracorporeal support was instituted emergently during LT for acute life-threatening conditions, not manageable with standard care [6].

The surgical technique for bilateral LT was consistent throughout the study period. The io ECMO positioning and management have been performed according to the technique previously published by Dell’Amore et al. and the criteria for cannulation have been clarified in Appendix A [19].

At the end of surgery, and in particular in recipients undergoing io VA ECMO, pulmonary function and hemodynamic parameters were monitored after chest closure. If they did not meet the predefined criteria for decannulation, peripheral VA ECMO was instituted (‘prolonged’ ECMO) [20].

The applied immunosuppressive regimen begins with an intravenous dose of methylprednisolone during the surgical procedure prior to each graft reperfusion, followed by an induction with basiliximab 20mg at postoperative day (POD) zero and four. The maintenance treatment consists of the following: (i) cyclosporine, with a target concentration of 250 ng/mL, from February 2016 to November 2019; or (ii) tacrolimus, with a target concentration of 10–15 ng/mL, after November 2019. Furthermore, both protocols include mycophenolate mofetil 1 to 2 g/day and methylprednisolone 0.5 mg/Kg/day [1,14,18,21].

The following variables were collected from electronic health records: (i) demographic data (age, gender, body mass index (BMI)); (ii) therapies at home (i.e., corticosteroids or O_2_-therapy); (iii) diabetes or chronic kidney injury; (iv) Oto score [22]; (v) lung-allocation score (LAS) [23]; (vi) underlying diseases leading to LT (see full description in Table 1); (vii) pre-existing recipient-related Gram-negative (GN) colonization; (viii) provenience (hospital, home); (ix) surgical characteristics (time of LT, time of graft ischemia, io fluid support and peri/po surgical revisions, bleeding needing surgery and thromboembolic/ischemic events); (x) io use of ‘prophylactic’, ‘rescue’ or ‘prolonged’ ECMO [1,5,6,20]; (xi) immunosuppressive therapy; (xii) length of invasive mechanical ventilation; (xiii) Clavien-Dindo score [24,25]; and (xiv) short- and mid-term outcomes of interest (72-h PGD), perioperative blood units (transfused within 72–96 h after LT), ICU length of stay (LOS), re-tracheal intubation and/or tracheostomy, AKI (only stage 2 or 3, according to the KDIGO guidelines) and/or renal replacement therapy, multi-drug resistant (MDR)/extended-beta lactamases (ESBL) gram-negative bacteria, acute cellular rejection within 30 days after LT (according to the International Society for Heart and Lung Transplantation criteria), and hospital (H) LOS and mortality [14,15,24,25,26,27].

### 2.2. Study Outcomes

Concerning the routine use of io VA ECMO: (i) primary outcome was 72-h PGD; and (ii) secondary outcomes were perioperative blood unit consumption, ICU and H LOS, re-tracheal intubation and/or tracheostomy, renal dysfunction and H mortality.

With regards to po immunosuppression with tacrolimus: (i) primary outcome was 30-day acute cellular rejection; and (ii) secondary outcomes were renal dysfunction, positive microbiology after surgery, re-tracheal intubation and/or tracheostomy, H mortality, ICU and hospital (H) length of stay (LOS).

### 2.3. Statistical Analysis

Descriptive statistics were reported as median/I quartile/III quartile for continuous variables, and absolute numbers (percentages) for categorical variables. The distribution of continuous and categorical variables was compared using Wilcoxon, Kruskal-Wallis and Pearson Chi-squared tests, respectively, according to the mode of io ECMO use and po tacrolimus.

To account for potential confounding related to the non-random allocation of the patients to the three groups of io support (‘off-pump’ vs. ‘prophylactic’ VA ECMO vs. ‘rescue’ VA ECMO) and to immunosuppressive therapy (tacrolimus vs. cyclosporine), propensity score weighting approaches were employed. Propensity scores were estimated using covariate balancing propensity score and a trimming of the weights was performed at 90° quantile [28,29]. According to clinical judgement, the variables considered for propensity scores estimation were gender, Oto score, LAS, BMI, provenience (hospital vs. home), and pre-existing recipient colonizations.

A weighted logistic regression analysis was adopted for binary outcomes. Results were reported as odds ratio (OR), 95% confidence interval (CI), and *p*-value. Weighted Gamma models were employed to assess the effect of the intervention on continuous outcomes since their distribution was found to be non-normal using Shapiro-Wilk. The marginal effect was computed considering the partial derivatives of the marginal expectation [30]. Results were reported as average marginal effect (AME), 95% CI, and *p*-value. *p*-values and CI of secondary outcomes underwent Bonferroni correction to account for false discovery rate.

Analyses were performed using R software within the packages rms, CBPS and WeightIt for propensity score weighting procedure estimation, and margins for AME computation [28,29,30,31,32].

## 3. Results

During the study period, a total of 153 consecutive patients undergoing bilateral LT were screened. After excluding 25 (16%) patients, 128 patients were deemed eligible for data analysis (Figure 1).

### 3.1. ‘ Off-Pump’-Group versus ‘Prophylactic’ or ‘Rescue’ VA ECMO

Demographic and clinical characteristics of the three groups, before and after LT, are listed in Table 1 and Table 2.

Overall, 47 patients (37%) were transplanted ‘off-pump’, 51 (40%) underwent io ‘prophylactic’ VA ECMO, and 30 (23%) required ‘rescue’ VA ECMO.

Compared to the ‘off-pump’-group, both the io ‘prophylactic’ VA ECMO and ‘rescue’ VA ECMO group showed preoperatively greater medicalizations (such as corticosteroids) and worse LAS (Table 1).

Moreover, compared to the other two groups, io ‘rescue’ ECMO patients recorded a longer duration of transplantation procedure, a greater need of fluid infusion, a higher incidence of ‘prolonged’ ECMO and postoperative surgical revisions, worse Clavien-Dindo scores, and longer invasive mechanical ventilation (Table 2).

The graphical distribution of unadjusted and adjusted (i.e., after propensity score weighting procedure) covariates is shown in Figure 2.

The resulting balancing was satisfying, except for LAS which was found to be not balanced after the weighting procedure. For this reason, all of the weighted regression models employed to evaluate the effect of VA ECMO on the outcomes of interest were further adjusted for the LAS to account for residual confounding.

After adjustment, compared to ‘off-pump’ treated patients, ‘prophylactic’ VA ECMO showed comparable 72-h PGD values, similar use of blood products and less renal dysfunction (Table 2).

On the contrary, ‘rescue’ VA-ECMO recorded a higher risk of 72-h PGD ≥ 2 and a greater need of perioperative blood transfusions (Table 2).

### 3.2. Tacrolimus versus Cyclosporine-Treated Patients

Demographic and clinical characteristics of the two groups are listed in Table 3 and Table 4. Forty-two (33%) patients were treated with tacrolimus, while eighty-six (67%) with cyclosporine.

As compared to the cyclosporine group, in the tacrolimus group preoperative corticosteroid therapies were less frequent and LAS was significantly greater (Table 3), while po surgical complications rated with lower Clavien-Dindo scores (Table 4).

The graphical distribution of unadjusted and adjusted pre-defined covariates and the final quality of balancing is shown in Figure 3. The balancing was satisfying. As compared to cyclosporine, the tacrolimus group showed a lower risk of 30-day acute cellular rejection. Secondly, a protective role was described also on the incidence of renal dysfunction and bacteria isolation (Table 4).

## 4. Discussion

This study, conducted during the last six years and enrolling consecutive adult recipients of bilateral LT, shows that the routine use of ‘prophylactic’ VA ECMO and the choice of tacrolimus are related to better LT short- and mid-term outcomes.

In the recent literature, the best io management of LT is still matter of debate, despite an increasing number of investigations reporting promising results in favour of a ‘routine’ use of VA ECMO compared to cardio-pulmonary bypass (CPB). In contrast, data are unclear about the potential superiority of io VA ECMO in relation to ‘off-pump’-group [3,24,33]. In fact, the use of CPB during LT has been abandoned due to its well-known adverse effects, such as the need for full heparinization, severe bleeding complications, higher po PGD values and more frequent renal dysfunction [24,34]. Moreover, CPB cannot be used out of the operating room [11].

Considering the ‘routine’ use of io VA ECMO compared to the ‘off-pump’-group, data are conflicting. Hoetzenecker and colleagues showed lower PGD values within 72 h after LT and superior short- and long-term survival in LT recipients undergoing ‘routine’ VA ECMO. No differences related to the respiratory support and the incidence of AKI were found [8,20]. Fessler and colleagues investigated a large cohort of LT patients, distinguishing between ‘off-pump’, ‘unplanned’-ECMO and ‘planned’-ECMO, as carried out in our investigation. Based on their findings, 72-h PGD, the incidence of re-intubation or tracheostomy and septic shock were comparable between ‘off-pump’ and ‘planned’-ECMO, while their data about the perioperative need of blood products were unclear [6]. In contrast, Ius and colleagues found a more complicated perioperative and early po course in recipients supported by VA ECMO versus ‘off-pump’, with no difference in PGD rates [33]. However, the authors did not perform any secondary analysis investigating potential differences between patients prophylactically treated with io VA ECMO versus those who were assisted later. Additionally, patients belonging to the ‘off-pump’-group, reported by the authors, exhibited lower pre-transplant surgical risk profiles (i.e., lower LAS and the presence of ‘bridge’ ECMO) making any comparison speculative [33,34]. Regarding the results of the largest international, multicenter registry analysis, ECMO was confirmed to be superior to CPB, but not compared to ‘off-pump’. Indeed, the ‘off-pump’ group seems to be associated with the lowest risk of severe PGD [24]. However, in the aforementioned registry, ‘prophylactic’ and ‘rescue’ ECMO were included in the same cohort and the protocols for the use of io extracorporeal support, anticoagulation and reperfusion strategy were not homogeneous between centers, leading to a relevant inter-center variability [7,24].

To the best of our knowledge, the present study is the first investigation in which the use of io VA ECMO is clearly distinguished between ‘prophylactic’ and ‘rescue’ and, excluding all recipients previously treated with ‘bridge’ ECMO or IMV, well-known biases. Indeed, a propensity score was applied for balancing preoperative clinical confounders, allowing us to obtain promising results in favour of ‘prophylactic’ use of io VA ECMO, which was not inferior to ‘off-pump’ treatment and superior to io ‘rescue’ VA ECMO [6,8,20,24,33]. Moreover, ‘prophylactic’ VA ECMO positively impacted the incidence of renal dysfunction [6,8,20]. Furthermore, the need for perioperative blood units was comparable to the ‘off-pump’ group, probably due to greater hemodynamic stability and consequently lower requirements for fluid support and blood products [7,35].

Considering the immunosuppressive protocol, our findings suggest that tacrolimus might reduce the occurrence of 30-day acute cellular rejection, the incidence of renal dysfunction and 30-day gram-negative bacteria isolation. These results have been only partially anticipated by Treede et al. and the Cochrane Review, published in 2013 and updated in 2018, showing a lower incidence of chronic lung allograft dysfunction in favour of tacrolimus while, considering the incidence of acute cellular rejection, data are still conflicting [13,14,21,36,37].

Moreover, two subsequent RCTs did not show a relevant impact of tacrolimus on infections and AKI but, in both investigations, no information has been provided about the occurrence of difficult-to-treat bacteria isolation, previous colonizations, the need for RRT, or baseline patient’s renal function [36,37]. In fact, our population was weighted for pre-existing colonization, a potential bias, and only two patients had pre-existing chronic renal dysfunction at stage 1. Indeed, we paid more attention to stages 2 and 3 of AKI since these levels of renal dysfunction have been shown to mostly affect LT survival [38].

About the risk of infection, we investigated the incidence of ESBL/MDR gram-negative bacteria isolation, as *Enterobacterales* and *Pseudomonas aeruginosa*, since they have been recognized to negatively impact on H LOS, short- and long-term survival and costs of hospitalization [1,17,18,39].

However, additional randomized studies are required to provide clear evidence about the benefit and safety of a routine use of io ‘prophylactic’ VA ECMO and po immunosuppressive regime based on tacrolimus among bilateral LT recipients.

Our study has some limitations. First, it is a retrospective observational study, suffering from the limits of designs of this kind. However, we used a propensity score weighting approach to avoid potential confounding related to the non-random allocation of the patients to different treatments. Second, the propensity score approach is a well-known method to account for confounding by indication. However, we cannot rule out that unmeasured or unknown variables potentially influencing the outcomes of interest could be left out. Third, we collected clinical data about cellular rejection within 30 days after LT, limiting our analysis only to short and mid-term outcomes. However, many studies have found a greater occurrence of comorbidities and acute cellular rejection within the first month following LT [2,21,40]. Four, the benefits related to a routine use of ‘prophylactic’ extracorporeal support, in some secondary outcomes, could be at least partially related to a mid-term administration of tacrolimus. Finally, we have considered only acute cellular rejection, and not antibody-mediated acute rejection, due to a clear standardization of diagnostic criteria only in recent years [40].

In conclusion, ‘prophylactic’ VA ECMO was associated with 72-h PGD values comparable to the ‘off-pump’-group; while po tacrolimus recorded a lower risk of 30-day cellular rejection and, secondly, less renal dysfunction and ‘difficult-to-treat’ bacteria isolations. 

## Figures and Tables

**Figure 1 jcm-11-05859-f001:**
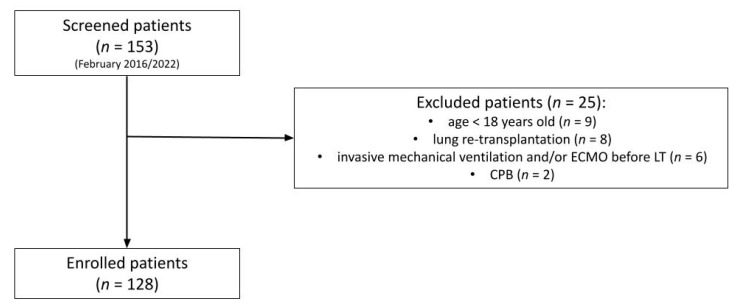
Flow chart of enrolled patients. Abbreviations: CPB, cardio-pulmonary bypass; ECMO, extracorporeal membrane oxygenation; LT, lung transplantation; *n*, number.

**Figure 2 jcm-11-05859-f002:**
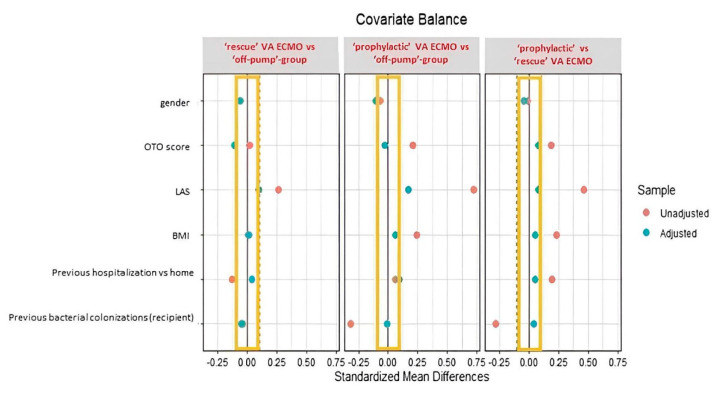
Propensity score (‘off-pump’ vs. ‘prophylactic’ VA ECMO versus ‘rescue’ device). The standardized mean differences before (red dots) and after (blue dots) the propensity score weighting procedure for the variables used in the propensity score estimation. Abbreviations: BMI, body mass index; LAS, lung allocation score; VA ECMO, veno-arterial extra-corporeal membrane oxygenation.

**Figure 3 jcm-11-05859-f003:**
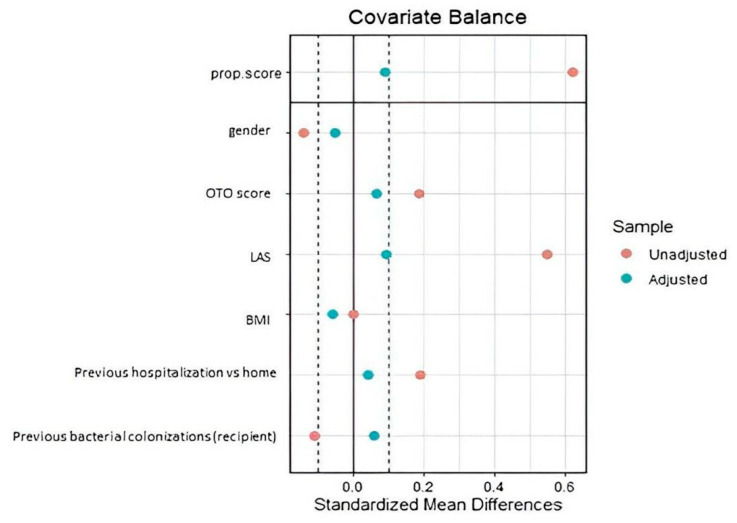
Propensity score (tacrolimus versus cyclosporine-group). The standardized mean differences before (red dots) and after (blue dots) the propensity score weighting procedure for the variables used in the propensity score estimation. Abbreviations: BMI, body mass index; LAS, lung allocation score.

**Table 1 jcm-11-05859-t001:** ‘Off-pump’-group, io ‘prophylactic’ VA ECMO, and ‘rescue’ device: baseline characteristics.

	‘Off-Pump’ N = 47 (37)	‘Prophylactic’ VA ECMO N = 51 (40)	‘Rescue’ VA ECMO N = 30 (23)	*p*-Value
**Baseline characteristics**				
Age, years	50 [43, 60]	56 [48, 61]	50 [40, 58]	0.201
M, n (%)	31 (66)	32 (63)	19 (63)	0.999
F, n (%)	16 (34)	19 (37)	11 (37)	
BMI, kg/m^2^	22 [20, 27]	24 [20, 28]	23 [21, 25]	0.550
Corticosteroids, n (%)	25 (53)	24 (47)	23 (77)	0.030
O_2_ therapy, n (%)	41 (87)	44 (86)	30 (100)	0.108
Diabetes, n (%)	6 (13)	12 (24)	5 (17)	0.421
Chronic renal disease, n (%)	1 (2)	0 (0)	1 (3)	0.474
Oto score	3 (1–5)	3 [2, 5]	3 [1, 5]	0.400
LAS	34 [33, 38]	37 [34, 44]	37 [34, 40]	0.003
**Underlying diseases ***				
Septic, n (%)	14 (30)	9 (18)	6 (20)	0.031
Interstitial, n (%)	17 (36)	33 (65)	15 (50)	
Obstructive, n (%)	10 (21)	9 (18)	4 (13)	
Others, n (%)	6 (13)	0 (0)	5 (17)	
**Previous colonization**				
Recipient-related, n (%)	18 (38)	12 (24)	11 (37)	0.200
None, n (%)	29 (62)	39 (76)	19 (63)	
**Provenience**				
Hospital, n (%)	3 (6)	4 (8)	1 (3)	0.900
Home, n (%)	44 (94)	47 (92)	29 (97)	

Data are expressed median [I-III quartile or confidential interval] for continuous variables and absolute numbers (percentage) for categorical ones. * Septic: cystic fibrosis, bronchiectasis; Interstitial: idiopathic pulmonary fibrosis, allergic extrinsic alveolitis, non-specific interstitial pneumonia, fibrosing emphysema, lymphocytic interstitial pneumonia, respiratory bronchiolitis interstitial lung; Obstructive: chronic obstructive pulmonary disease, emphysema; Others: idiopathic pulmonary hypertension, veno-occlusive disease, connective tissue disease, α1-anti-tripsin deficiency, lymphangioleiomyomatosis, histiocytosis, sarcoidosis, graft versus host disease. Abbreviations: BMI, body mass index; F, female; io, intraoperative; LAS, lung allocation score; LT, lung transplantation; M, male; n, number.

**Table 2 jcm-11-05859-t002:** ‘Off-pump’-group, ‘prophylactic’ VA ECMO and ‘rescue’ device: peri- and postoperative characteristics and outcomes.

	‘Off-Pump’ N = 47 (37)	‘Prophylactic’ VA ECMO N = 51 (40)	‘Rescue’ VA ECMO N = 30 (23)	*p*-Value	
**Surgical characteristics**					
Time of LT, minutes	420 [370, 470]	413 [371, 450]	490 [450, 533]	0.001	
Time of graft ischemia, minutes	565 [460, 630]	585 [473, 678]	573 [480, 715]	0.662	
Io fluid support, mL	4200 [3400, 5175]	4250 [2925, 5640]	5600 [4050, 7850]	0.010	
‘*prolonged*’ ECMO, n (%)	0 (0)	11 (22)	12 (40)	0.001	
**During hospitalization**					
Surgical revisions, n (%)	6 (13)	10 (20)	11 (37)	0.041	
Surgical bleeding, n (%)	2 (4)	5 (10)	6 (20)	0.083	
Thromboembolic/ischemic events, n (%)	2 (4)	1 (2)	2 (7)	0.570	
Invasive mechanical ventilation, hours	22 [18, 36]	47 [24, 154]	60 [36, 434]	0.001	
Clavien-Dindo score	21 [0, 46]	0 [0, 42]	47 [5, 61]	0.001	
Bacteria isolation, n (%)	23 (49)	13 (25)	11 (37)	0.055	
30-day acute cellular rejection *, n (%)	14 (31)	7 (16)	4 (17)	0.083	
**Primary outcome**				**OR a, *p*-value**	**OR b, *p*-value**
PGD at 72 h (ref. PGD ≥ 2)	1 [0, 2] 14 (37)	0 [0, 2] 16 (33)	2 [1, 3] 19 (68)	0.69 [0.39, 1.24], 0.210	3.44 [1.94, 6.10], 0.001
**Secondary outcomes**				**AME a, *p*-value**	**AME b, *p*-value**
Perioperative blood units **, n	2 [1, 3]	4 [2, 6]	8 [4, 14]	1.08 [−1.72, 3.89], 1	8.12 [2.22–14.03], 0.001
ICU LOS, days	7 [5, 13]	8 [5, 14]	17 [9, 32]	−3.80 [−12.18, 4.59], 1	9.07 [−3.22, 21.35], 0.414
H LOS, days (%)	32 [28, 44]	33 [29, 43]	38 [31, 48]	−6.63 [−22.47, 9.20], 1	4.08 [−13.34, 21.49], 1
				**OR *a, p*-value**	**OR *b*, *p*-value**
Re-intubation and/or tracheostomy, n (%)	11 (23)	13 (25)	13 (43)	0.63 [0.25, 1.55], 1	1.92 [0.87, 4.37], 0.235
Renal dysfunction, n (%)	13 (28)	6 (12)	15 (30)	0.19 [0.06, 0.53], 1	2.05 [0.95, 4.52], 0.098
H mortality, n (%)	4 (9)	4 (8)	7 (23)	0.33 [0.06, 1.43], 0.472	2.46 [0.88, 7.70], 0.189

Data are expressed median [I-III quartile] for continuous variables and absolute numbers (percentage) for categorical ones. a: ‘off-pump’-group vs. io ‘prophylactic’ VA ECMO; b: ’off-pump’-group vs. ‘rescue’ device (weighted Gamma models were employed to assess the effect of the intervention on continuous outcomes. A weighted logistic regression approach was adopted for binary outcomes). Confidential interval was reported [CI]. *: rejection is defined according to International Society for heart and lung transplantation (ISHLT) criteria (i.e., A3–A4 and/or B2 grade at biopsy) [18]. **: anticoagulation administration and blood transfusions were performed according to current evidences [5,12,18,19,20]. Abbreviations: AME, average marginal effect; H, hospital; ICU, intensive care unit; io, intraoperative; LOS, length of stay; LT, lung transplant; n, number; OR, odds ratio; V-A ECMO, veno-arterial extracorporeal membrane oxygenation.

**Table 3 jcm-11-05859-t003:** Tacrolimus- vs. cyclosporine-treated group: baseline characteristics.

	Overall N = 128 (100)	Tacrolimus N = 42 (33)	Cyclosporine N = 86 (67)	*p*-value
**Baseline characteristics**				
Age, years	53 [43, 60]	55 [42, 61]	52 [44, 60]	0.401
M, n (%)	82 (64)	25 (60)	57 (66)	0.502
F, n (%)	46 (36)	17 (40)	29 (34)	
BMI, kg/m^2^	23 [20, 27]	24 [20, 27]	23 [21, 27]	0.999
Corticosteroids, n (%)	72 (56)	18 (43)	54 (63)	0.040
O_2_ therapy, n (%)	115 (90)	36 (86)	79 (92)	0.350
Diabetes, n (%)	23 (18)	10 (24)	13 (15)	0.230
Chronic renal disease, n (%)	2 (2)	0 (0)	2 (2)	0.999
Oto score	3 [1, 5]	3 [2, 5]	3 [1, 5]	0.201
LAS	36 [33, 40]	38 [34, 45]	35 [33, 39]	<0.001
**Underlying diseases ***				
Septic, n (%)	29 (23)	9 (21)	20 (23)	0.010
Interstitial, n (%)	65 (51)	26 (62)	39 (45)	
Obstructive, n (%)	23 (18)	7 (17)	16 (19)	
Others, n (%)	11 (9)	0 (0)	11 (13)	
**Previous colonization**				
Recipient-related, n (%)	41 (32)	12 (29)	29 (34)	0.600
None, n (%)	87 (68)	30 (71)	57 (66)	
**Provenience**				
Hospital, n (%)	8 (6)	4 (10)	4 (5)	
Home, n (%)	120 (94)	38 (90)	82 (95)	0.400

Data are expressed median [I-III quartile] for continuous variables and absolute numbers (percentage) for categorical ones. * Septic: cystic fibrosis, bronchiectasis; Interstitial: idiopathic pulmonary fibrosis, allergic extrinsic alveolitis, non-specific interstitial pneumonia, fibrosing emphysema, lymphocytic interstitial pneumonia, respiratory bronchiolitis interstitial lung; Obstructive: chronic obstructive pulmonary disease, emphysema; Others: idiopathic pulmonary hypertension, veno-occlusive disease, connective tissue disease, α1-anti-tripsin deficiency, lymphangioleiomyomatosis, histiocytosis, sarcoidosis, graft versus host disease. Abbreviations: BMI, body mass index; F, female; LAS, lung allocation score; LT, lung transplantation; M, male; n, number.

**Table 4 jcm-11-05859-t004:** Tacrolimus- vs. cyclosporine-treated group: peri- and post-operative, characteristics and outcomes.

	Overall N = 128 (100)	Tacrolimus N = 42 (33)	Cyclosporine N = 86 (67)	*p*-Value
**Surgical characteristics**				
Time of LT, minutes	435 [376, 484]	430 [375, 490]	448 [405, 498]	0.010
Time of graft ischemia, minutes	568 [480, 655]	580 [510, 655]	568 [480, 660]	0.980
Surgical revisions, n (%)	27 (21)	5 (12)	22 (26)	0.110
Surgical bleeding, n (%)	13 (10)	3 (7)	10 (12)	0.541
**Primary outcomes**				**OR a, *p*-Value**
30-day acute cellular rejection *, n (%)	25 (22)	3 (8)	22 (29)	0.21 [0.09, 0.48], 0.010
**Secondary outcomes**				**OR a, *p*-Value**
Re-tracheal intubation and/or tracheostomy, n (%)	37 (29)	8 (19)	29 (34)	0.33 [0.14, 0.73], 0.002
Renal dysfunction, n (%)	34 (27)	3 (7)	31 (36)	0.10 [0.03, 0.28], 0.001
Bacteria isolation, n (%)	47 (37)	9 (21)	38 (44)	0.41 [0.19, 0.85], 0.009
H mortality, n (%)	15 (12)	1 (2)	14 (16)	0.04 [0.01, 0.28], 0.008
				**AME b, *p*-Value**
ICU LOS, days	9 [6, 18]	7 [5, 14]	10 [6, 22]	−8.07 [−14.56, −1.57], 0.006
H LOS, days	33 [28, 46]	33 [30, 44]	33 [28, 46]	−7.14 [−18.11, 4.30], 0.62

Data are expressed median [I-III quartile] for continuous variables and absolute numbers (percentage) for categorical ones. a: A weighted logistic regression approach was adopted for binary outcomes; b: Weighted Gamma models were employed to assess the effect of the intervention on continuous outcomes. Confidential interval was reported [CI]. *: rejection is defined according to International Society for heart and lung transplantation (ISHLT) criteria (i.e., A3–A4 and/or B2 grade at biopsy) [18]. Abbreviations: AME, average marginal effect; H, hospital; ICU, intensive care unit; io, intra-operative; LOS, length of stay; LT, lung transplant; MDR/ESBL, multi-drug resistant/extended beta-lactamase; n, number; OR, odds ratio; V-A ECMO, veno-arterial extracorporeal membrane oxygenation.

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
