# Peer review of "The Impact of New Treatments on Short- and MID-Term Outcomes in Bilateral Lung Transplant: A Propensity Score Study"

_jcm, 2022, doi:10.3390/jcm11195859_

Round 1
Reviewer 1 Report
Authors did good job in writing this manuscript. They tried to study short and mid term outcomes of intraoperative VA ECMO and tacrolimus in bilateral lung transplant patients. They found that prophylactic intraoperative VA ECMO is associated with comparable 72hr PGD values compared to off pump group and tacrolimus is associated with lower risk of 30 day cellular rejection.
H mortality. 33: Please use full form atleast for the first time in manuscript and then you can use H mortality.
owed by an induction with basiliximab 20mg/die at postoperative day (POD) 0 and 4. 99: correct units for basiliximab
H mortality, ICU and H LOS. 138: Same as mentioned above; Use full form for H LOS for first time use in manuscript and then use abbreviated form.
CONCLUSIONS 256: change to discussion and if you like you can add conclusion at end if manuscript
debat, despite 260
immunosuppressive regime based on tacrolimus among bilateral LT recipients 320
Reviewer 2 Report
I have read the manuscript with interest, where the authors propose to evaluate the effect of the immunosuppression regimen and the prophylactic use of ECMO on post-transplant outcomes.
The manuscript is interesting and covers a controversial topic. However, it has certain limitations and aspects that need to be improved.
First of all, The prophylactic ECMO group used tacrolimus in most cases. We cannot distinguish whether the positive effects are secondary to the use of ECMO or the use of tacrolimus.
Secondly, Using the Propensity Score is central, to ensure that the groups are comparable. I suggest detailing more than propensity score modality was used. Multivariate modeling, matching, or stratification ??
Third, regardless of the propensity score, there is a risk of bias by indication or confounding by indication. Because this model only adjusts for some variables, unmeasured or unknown variables are left out. This should be mentioned in the limitations.
Fourth, underlying diseases are different between groups. This can modify the results. Is known that one of the main determinants of Long term outcomes is the underlying disease. Do you think that this imbalance can affect the results? Justify.
Fifth, It should be known if, in the timeline, ECMO has been used prophylactically in recent years. Because if there is a temporary difference in the use of prophylactic and rescue ECMO, part of the differences found may be due to the learning curve.
Six, In your center, did mortality and transplant complications change over time? In what time (over the years) were prophylactic ECMO and rescue ECMO used?
Minor comments:
- Table 2 is misaligned
- The figures 2 are very small and it is difficult to read them
- In the flowchart, draw monolateral LT.
- There are too many acronyms that make reading hard.
Reviewer 3 Report
The paper is interesting, but some changes are required. A linguistic edit is needed, and the number of abbreviations in the Abstract has to go down. Please consider using only a few or no abbreviations there, as they require much time to find out what they are. It is better to have a shorter and streamlined Abstract. I am not sure I understand the numeric presentations in the Abstract either. Can you please explain this better and re-phrase it to increase readability? I think that the writing style needs to be easier and more generalized, rather than somewhat restricted as it is now. For example, the aim sentence does not bring in much in the first reading; it takes a lot of re-reading to understand it. There is a fundamental risk of confounding by indication, which stems from the different entry statuses of the patients. This is reflected in Table 1, where we see the significant differences across the groups. Can you show that these groups are somehow poolable into the same analysis? Otherwise, it might be arguedd that the severity defined all of the outcome, and then it was masked by other effects. Table 1 has to be clarified more, what are the numbers in the brackets? Square brackets? Why are the groups so difficultly named, couldn't those be simplified? Please provide three decimal digits for P. Why use weighted regression, and weighted by what and how? Please do not refere to the clinical judgment in the variable selection, this should ideally be assessed in a better way. Arguably, you have so many tests that you should consider a false-discovery rate adjustment.
Round 2
Reviewer 2 Report
Comments made by reviewers have been adequately answered. The modifications made improved the quality of the manuscript. The article has certain limitations but can be published.
Reviewer 3 Report
Thank you for a detailed response